# Analysis of Sea Pollution by Sewage from Vessels

**Žarko Koboević** *, **Darijo Mišković** , **Romana Capor Hrošik and Nikša Koboević**

Maritime Department, University of Dubrovnik, Ćira Carića 4, 20000 Dubrovnik, Croatia;
darijo.miskovic@unidu.hr (D.M.); rcapor@unidu.hr (R.C.H.); niksa.koboevic@unidu.hr (N.K.)
* Correspondence: zarko.koboevic@unidu.hr

**Abstract:** In this study, we analysed the sea pollution caused by sewage from vessels. The Dubrovnik aquatorium was chosen as a typical sea area that accommodates a variety of vessels in different locations. We sampled the sea at eight different coastal locations over 14 months and then analysed the samples to determine the presence of the indicators of fecal pollution. Simultaneous with the sampling of the sea, we recorded the number and type of vessels accommodated at the port. These data were applied in chi-square tests, which were used to determine the existence of the relationship of certain types of vessels with fecal coliform bacteria in the sea for each location. The correlation was determined between smaller vessels such as boats, yachts, megayachts, and smaller cruise ships in national navigation with bacteria at sea at the sampling locations. The results can provide an improved understanding of sea pollution due to sewage from vessels.

**Keywords:** vessels; fecal coliform bacteria; sampling; chi-square test; risk assessment

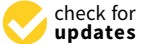



## 1. Introduction

Global population growth and increased trade and tourism have led to increased use of recreational and tourism-based boating activities [1,2]. Daily shipping and boating operations or practices, associated with accidental waste discharges/occurrences, all contribute to environmental pollution impacts [3,4], which can be direct or indirect depending on factors such as intensity and frequency (e.g., cumulative impacts of grouped vessels), and movement and specific activities of ships and boats based on daily routines and environmental conditions [5]. Given the increases in the number of vessels and their passengers, the risk of marine pollution with various substances has increased, especially along the coast, which includes sewage wastewater from vessels [6]. Sources of sewage from ships and boats include toilets being flushed directly into the receiving waters or discharge from a vessel's sewage treatment systems or holding tanks [2]. This source of sewage is considered to be most problematic within enclosed inland waters and/or semi-enclosed coastal waters with minimal flushing. National parks are particularly sensitive to sewage waters (e.g., marine parks), as are areas that continuously experience a large volume of vessels with a relatively large number of people on board, or where human activities are associated with primary human contact (e.g., swimming, fishing, and diving) [7].

Vessel-sourced sewage discharge can occur either at low continuous rates (e.g., direct releases from onboard toilets) or via large peaks (e.g., pump-outs of holding tanks while at sea). The volume of sewage discharged by a particular vessel is a product of passenger numbers and the onboard sewage management equipment rather than the overall vessel size [8]. Pollution of ports is determined by fecal indicator bacteria (FIB). The spatio-temporal dynamics of FIB were assessed in 12 ports of the Adriatic Sea. FIB were abundant in ports, often more so than in adjacent areas; their abundance patterns were related to salinity, oxygen, and nutrient levels. However, differences between the in-port and out-of-port samples varied. One reason for the differences found between these values may be explained by, in some cases, the stations outside the port being quite far from potential fecal sources, whereas in others, they were close to the city centre, which is a

source of significant FIB inputs [9]. The consequences of marine pollution by sewage can be permanent and have a major impact on the living marine environment, society, and economy. Sewage water consists of organic matter, nutrients (nitrogen, phosphorus, and potassium), inorganic matter (dissolved minerals), toxic chemicals (heavy metal and pesticides), and pathogens [10]. Contamination of coastal water may result in changes in nutrient levels; the abundance, biomass, and diversity of organisms; the bioaccumulation of organic and inorganic compounds; and alterations in the trophic interactions among species. Receiving waters with high flushing capacity are able to dilute or eliminate most of the conventional pollutants, but persistent toxic compounds and long-lived pathogens are posing management challenges [11].

The impact of vessel sewage wastewater on the receiving marine environment related to navigation is detrimental to the receiving environment and must be identified and effectively managed to minimize adverse effects, wherever possible, through the use of direct and indirect management tools, strategies, and techniques (e.g., education, legislation, local authorities, selective use of alternative technology, constraints, and management guidelines) to ensure the long-term sustainability of these environments. Sustainability, as a generally accepted concept, includes management to ensure that future generations preserve resources and inherit a clean environment [10]. Given the potential environmental impacts on coastal seas due to the discharge of sewage from a vessel, many countries are implementing legal requirements to pump untreated sewage from their sewage holding tanks into onshore wastewater treatment infrastructure. In many cases, to achieve this, vessels must use onshore stationary sewage receiving devices or pump the contents of sewage tanks into trucks or barges, or alternatively be served by pump-out boats moving in and between marinas to collect sewage from other vessels [12].

Since the early 2000s, the Adriatic Sea has experienced an increase in maritime traffic due to nautical and cruise tourism. Hence, there are more boats, yachts, and smaller and larger cruise ships, both in international and national navigation.

The authors of this study have a strong motivation to improve the current state of marine pollution by sewage from vessels in the Republic of Croatia. The causes for inadequate efficacy in sea protection from vessels sewage are numerous and complex. The most common reasons are inability to sail off the coast where dumping is permitted, a lack of reception facilities for sewage water in local aquatoriums, lack of adequate legal constraint, lack of responsible ecological awareness, etc. In addition, the volume of tanks for sewage water on smaller vessels is very limited, and when they are normally used, they quickly fill up and often need to be emptied.

The current situation in Croatian coastal sea can be improved only if changes are made in several areas. Improving legislation is the basis for the next steps. Most of the regulations concerning marine pollution from ships in Croatia are the result of adopting relevant international conventions under the International Maritime Organisation, but also of the implementation of the European Union rules and regulations. The Republic of Croatia has adopted the MARPOL Convention. National regulations should apply to the specific pollution problems not covered by the international regulations or in cases the adopted measures are not efficient enough to offer a quality protection from shipboard sewage pollution. There are countries that have dealt with the similar problem of sewage pollution from ships. They have found solutions in adoption of their national regulations or in an even more efficient combination of international and national regulations. Such positive experiences and solutions might simply be adjusted to the specific features of the Adriatic Sea and Croatian legislation.

Secondly, it is necessary to establish and build a sufficient number of easily accessible systems of receiving infrastructure for sewage from vessels. Finally, the third step is to establish a system of control over the use of available reception systems or sanctioning violators for uncontrolled discharge of sewage water from vessels into the surrounding sea. First of all, researchers need to provide scientific evidence that the sea is contaminated with sewage from vessels in an area so that they can propose the adoption of new regula-

tions, installation of reception infrastructure for sewage from onshore vessels and control their use.

A typical example of an area with increasing maritime traffic of smaller recreational vessels, but also the largest cruise ships as part of travel itinerary, is the city of Dubrovnik, as one of the most important destinations on the Adriatic and the Mediterranean Seas.

There are several locations in the Dubrovnik aquatorium where different types of vessels reside, depending on their activities. Therefore, we selected the following locations where different types of vessels most often reside for sea sampling, presented in Figure 1:

1.   ACI marina Dubrovnik (mooring for yachts and boats);
2.   Cavtat, the waterfront area (mooring for yachts and megayachts);
3.   Gruž Harbour, international port for passenger traffic (berths for cruise ships);
4.   Gruž waterfront, operational waterfront (berths for yachts, smaller cruisers in national navigation and excursion boats);
5.   The island of Lopud, anchorage in Lopud (anchorage for boats and yachts between the waterfront and the beach);
6.   The island of Lopud, Šunj Bay (anchorage in front of the beach for boats, yachts, megayachts, and smaller cruisers in national navigation);
7.   The island of Lokrum, anchorage in front of the Dubrovnik old town port (anchorage for cruises and megayachts);
8.   Zaton, anchorage in the bay, in front of the village of Zaton, anchorage for boats.

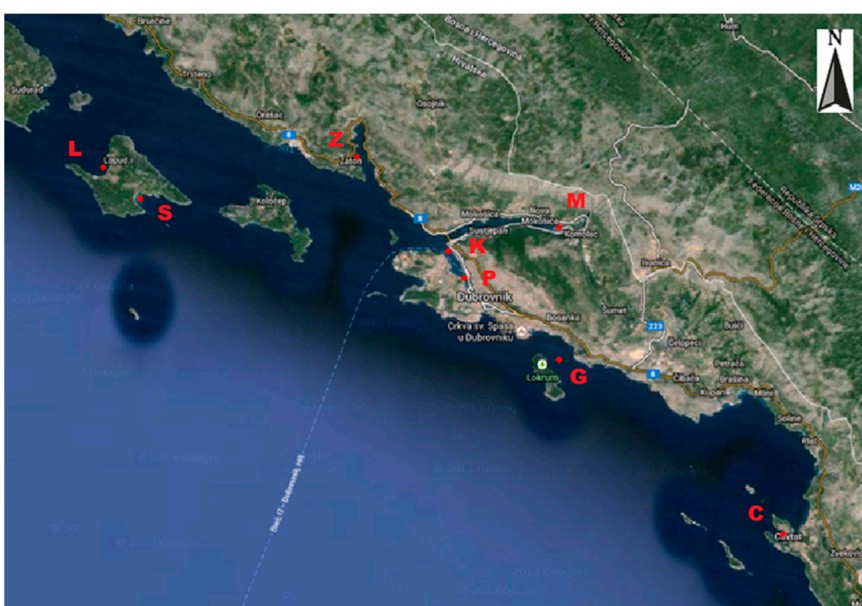

**Figure 1.** Sea sampling locations in the Dubrovnik aquatorium. (C, Cavtat waterfront; G, Dubrovnik old town anchorage; K, Gruž harbour; L, Lopud; M, ACI marina Dubrovnik; P, Gruž waterfront; S, Šunj; Z, Zaton).

## 2. Materials and Methods

For risk verification, we chose the statistical data processing method, in which we compared the indicator of the fecal load of the sea from seawater samples with the type and number of vessels, or the number of persons on board at the time of sampling at a particular sea sampling location. Thus, we verified the correlations and the trends.

Fecal coliform bacteria were used as indicators of marine fecal load because they are one of the key indicators that a vessel's black wastewater treatment plants must meet before receiving a "class register certificate". In addition, the MARPOL Convention in Annex IV prescribes the limits with respect to the amount of fecal coliform bacteria in the effluent to be discharged into the sea.

As for fecal contamination testing, the fecal coliform test is more specific than the total coliform test because fecal coliforms are used as standard indicators in many laboratories to test treated sewage, public water supply, and primary contact water such as swimming areas [13]. Identifying the presence of fecal pathogens and the contaminant source in bathing areas is important for obtaining microbial infection risk estimates, preventing contamination events, and protecting public health [14,15].

Microbiological analysis of seawater samples was performed by an authorized and specialized institution in the Dubrovnik-Neretva County (Public Health Institute of the Dubrovnik-Neretva County, Health Ecology Service) using the membrane filtration method.

The seawater was sampled in accordance with Article 15 of the Regulation on the quality of bathing sea (Official Gazette No. 73/08) and Annex II. Rules on sample handling for microbiological analysis, which adopts the standards from the EU Directive 2006/07/EC of the European Parliament and the Council of 15 February 2006 concerning the management of bathing water quality and repealing Directive 76/160/EEC [16], which are also described in ISO7899-1 and ISO 7899-2 standards. Thus, sampling was always performed at the same sampling point at a particular location with a manual sampler, and the bottle was marked with a code (consisting of a code letter for the location and a number for the date). The sampler was immersed at a depth of 1 m below the sea surface. Sea samples were not taken during heavy rain, strong wind, large waves or the occurrence of macroalgal/phytoplankton proliferation.

Legal limits for fecal coliforms in bathing waters are set by EU Directive 2006/07/EC of the European Parliament and the Council of 15 February 2006 concerning the management of bathing water quality and repealing Directive 76/160/EEC, as also described in ISO7899-1 and ISO 7899-2 standards. The limits for fecal coliforms for coastal waters and transitional waters are excellent quality $\leq 100$, good quality $\leq 200$, sufficient $\leq 185$ CFU/100 mL [17].

Simultaneously with the sea sampling, we counted the vessels and recorded the number of people (best estimated number (BEN)) spending time on these vessels in locations where no records from relevant institutions were available (locations: Lopud, Šunj, Zaton, Cavtat, Gruž-riva, and the Dubrovnik Old City anchorage). In case of locations where such data existed, they were obtained from the appropriate institutions, such as the Port of Dubrovnik Authority and the ACI Marina Dubrovnik for the locations of Gruž harbour, ACI Marina Dubrovnik, and Dubrovnik Old City anchorage.

The results of the marine samples, i.e., the number of fecal coliform bacteria at certain sampling points (locations), had to be processed together with the number of vessels, their type or the number of persons on them in order to determine whether there was mutual correlation between the number of fecal coliform bacteria found in water samples and the vessel types and/or the number of persons on them.

For this data processing, the statistical method we used was the chi-squared test ($\chi^2$ test), which is used for testing correlations between two qualitative (categorical) variables, and/or for deviations in the obtained frequencies of occurrence of one variable from those expected [18]. In almost all examples, the chi-square is calculated in the same way, according to the formula:

$$\chi^2 = \sum \frac{(f_0 - f_t)^2}{f_t} \tag{1}$$

where $f_0$ is the observed frequencies, and $f_t$ is the expected (theoretical) frequencies, i.e., the ones we would expect with some particular hypothesis. The sum of the observed frequencies must be equal to the sum of the expected frequencies [19].

Research starts with an assumption of a null hypothesis (H0) that there is no statistically significant difference between the observed and expected frequency distributions, i.e., that it occurred by chance. If there is no difference, then $x^2 = 0$. The larger the differences between the observed and expected frequencies, the larger the definitive expression $\chi^2$. Therefore, the smaller the chi-square, the more likely the hypothesis is to be accepted, and the larger it is, the more likely the hypothesis should be rejected because the observed results differ significantly from those we would expect.

The critical value (cv) is the value of the test for which the null hypothesis is rejected. By entering the degree of freedom (v) and the significance level ($\alpha$) in the table, and using a degree of freedom of 1 and a significance level of 0.05, we obtained the critical value (cv) from the table, 3.84 [20]. Thus, for each calculated value of the chi-square that was more than 3.84, we concluded that the set null hypothesis should be rejected because the probability that the null hypothesis is true was less than 95%.

## 3. Results and Discussion

### 3.1. Sea Sampling Results at Locations

Sea sampling was carried out for 14 consecutive months during all seasons, covering two tourist seasons. It started in August 2012 and ended in October 2013, including winter when there were no vessels and during the summer months when there were vessels at the sampling points. No sewage contamination indicators (fecal coliform (FC)) appeared at any of the sites when there were no vessels at the sampling site, but we note that people live permanently in nearby houses year-round. This suggests that there is no inflow of sewage water from nearby houses into the sea because if sewage entered the sea from nearby houses, it would be detected all months of the year.

The results of seawater sample analysis conducted by the Health Ecology Service of the Public Health Institute of Dubrovnik-Neretva County by location (sampling point) and by month of sampling are presented in Table 1.

**Table 1.** Number of fecal coliform bacteria in CFU/100 mL at locations by months sampling.

| Sampling Point | M | C | K | P | L | S | G | Z |
|---|---|---|---|---|---|---|---|---|
| **Month of Sampling** | **ACI Marina** | **Cavtat Pier** | **Gruž Harbour** | **Gruž Pier** | **Lopud** | **Šunj** | **Old Town Anch.** | **Zaton** |
| Sep. 2012 | 223 | 1290 | 0 | 148 | 0 | 0 | 4 | 62 |
| Oct. 2012 | 40 | 0 | 0 | 15 | 0 | 0 | 5 | 0 |
| Nov. 2012 | 20 | 0 | 2 | 308 | 2 | 0 | 0 | 0 |
| Dec. 2012 | 34 | 0 | 78 | 6 | 0 | 0 | 0 | 0 |
| Jan. 2013 | 168 | 0 | 9 | 3 | 0 | 0 | 0 | 0 |
| Feb. 2013 | 23 | 0 | 44 | 65 | 0 | 0 | 0 | 1 |
| Mar. 2013 | 22 | 4 | 239 | 1920 | 0 | 0 | 0 | 0 |
| Apr. 2013 | 8 | 0 | 0 | 550 | 0 | 0 | 0 | 0 |
| May 2013 | 0 | 0 | 2 | 0 | 0 | 1 | 0 | 1 |
| Jun. 2013 | 78 | 28 | 0 | 2 | 0 | 0 | 0 | 0 |
| Jul. 2013 | 190 | 145 | 0 | 0 | 700 | 74 | 0 | 0 |
| Aug. 2013 | 97 | 1240 | 0 | 0 | 1760 | 0 | 0 | 0 |
| Sep. 2013 | 290 | 6 | 0 | 0 | 0 | 31 | 0 | 0 |
| Oct. 2013 | 0 | 2 | 0 | 0 | 0 | 0 | 0 | 0 |

The distribution of fecal coliform bacteria, vessels by type at the time of sea sampling, and the best estimated number of persons on these vessels are shown in the Table 1.

### 3.1.1. Location and Sampling Results in ACI Marina Dubrovnik

In the ACI Marina Dubrovnik, the sampling was performed between berths 2 m from the pier, (Figure 2) always at the same location for 14 consecutive months.

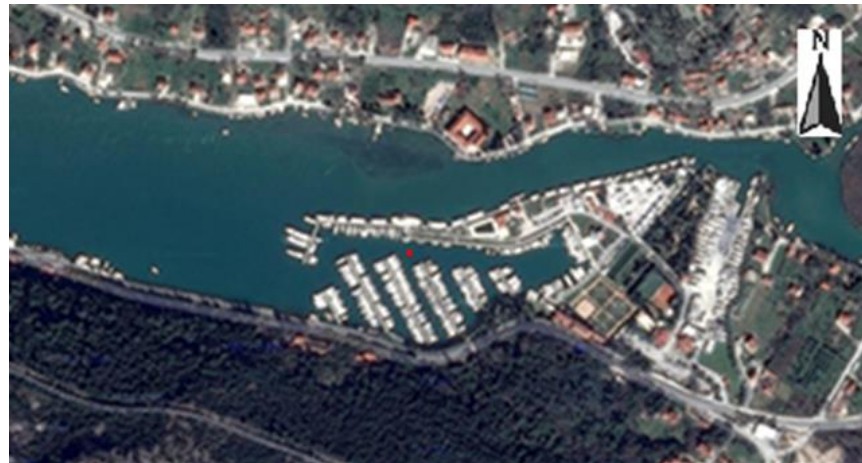

**Figure 2.** Satellite image of ACI Marina Dubrovnik (the red dot shows the sampling point).

The marina is located in the fjord near the source of the Ombla River. That is why there are strong sea currents along the northern berths of the marina (toward the middle of the river). These sea currents carry impurities (including sewage discharged from vessels) from the marina to the river mouth. Thus, sea currents clear the marina of sewage from the vessels. In the inner part of the marina, at the sampling point, the water flow is slower but constant. Therefore, we concluded that the number of fecal coliforms would be significantly higher in still water with the same discharge of black wastewater from vessels (yachts and boats) into the surrounding sea while staying at the marina.

Table 2 shows that we found slightly fewer vessels (not a significant number) in the marina during the winter months. However, the number of people on vessels in the summer months was significantly higher. Although there were no registered guests (persons) at the marina reception during the three winter months, a number of people do not have to register, such as Croatian and other EU citizens. In addition, a number of people work on the maintenance of boats and yachts during the day. All of these persons use toilets on the vessels, and the contents are occasionally discharged into the surrounding sea while berthed. The data from Table 2 are graphically shown in Figure 3.

**Table 2.** Distribution of vessels and persons on board vessels on sampling days at site M, ACI Marina Dubrovnik.

| | Sampling Point M, ACI Marina (Boats and Yachts) | | | |
|---|---|---|---|---|
| **Sampling Date** | **Number of Bacteria CFU/100 mL** | **Type of Vessels at Sampling Location** | **Number of Vessels** | **Most Probable No.of Persons** |
| 12 Sep. 2012 | 223 | boats and yachts | 399 | 648 |
| 18 Oct. 2012 | 40 | boats and yachts | 371 | 308 |
| 18 Nov. 2012 | 20 | boats and yachts | 354 | 24 |
| 20 Dec. 2012 | 34 | boats and yachts | 357 | 0 |
| 28 Jan. 2013 | 168 | boats and yachts | 355 | 0 |
| 25 Feb. 2013 | 23 | boats and yachts | 360 | 0 |
| 24 Mar. 2013 | 22 | boats and yachts | 369 | 51 |
| 14 Apr. 2013 | 8 | boats and yachts | 371 | 89 |
| 21 May 2013 | 0 | boats and yachts | 393 | 93 |
| 26 Jun. 2013 | 78 | boats and yachts | 388 | 334 |
| 15 Jul. 2013 | 190 | boats and yachts | 385 | 721 |
| 16 Aug. 2013 | 97 | boats and yachts | 437 | 781 |
| 9 Sep. 2013 | 290 | boats and yachts | 387 | 620 |
| 20 Oct. 2013 | 0 | boats and yachts | 361 | 312 |

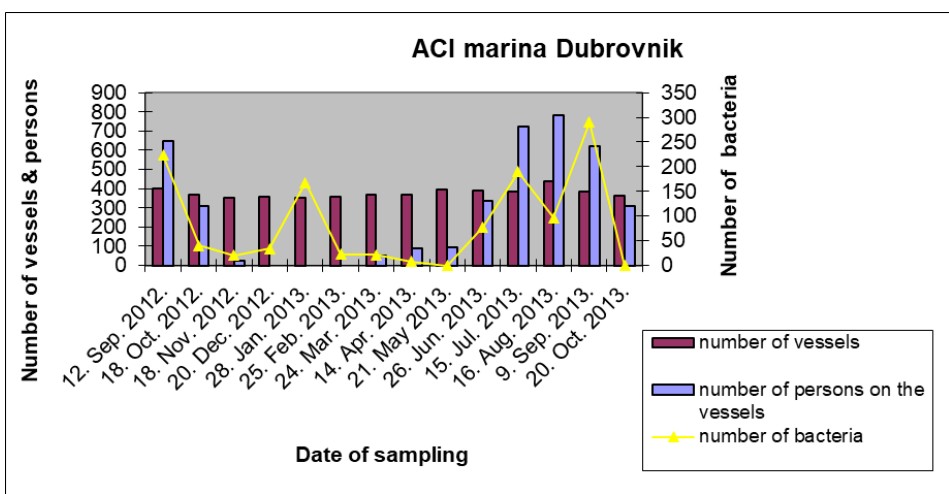

**Figure 3.** The number of fecal coliform bacteria, vessels, and persons on board at the sampling location at the ACI Marina Dubrovnik.

3.1.2. Location and Sampling Results in the Cavtat Waterfront

The Cavtat waterfront is a popular mooring for yachts and megayachts during the summer months because of its attractiveness for boaters, and because of its geographical position shown in Figure 4.

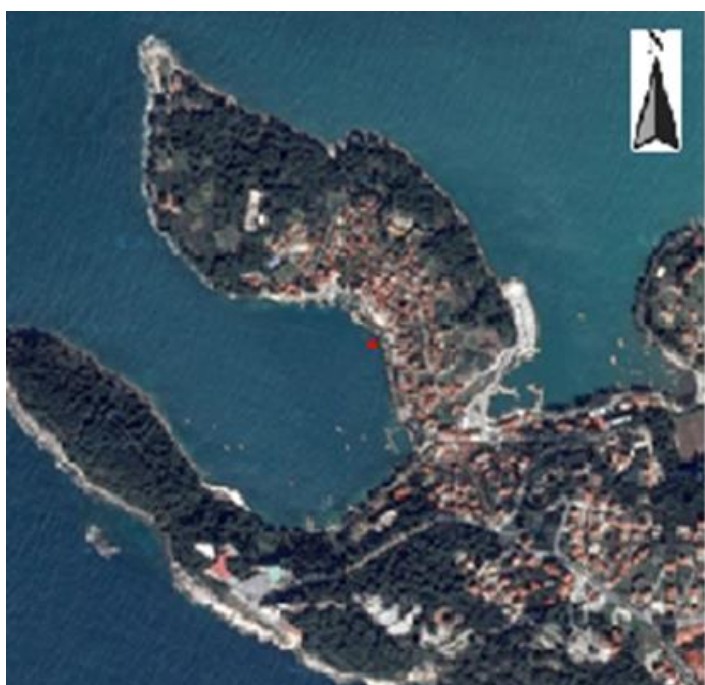

**Figure 4.** Satellite photo of the Cavtat waterfront (red dot shows sampling point).

All foreign vessels that sail along the Adriatic coast and enter the territorial waters of Croatia, and come from the territorial waters of Montenegro, are required to report to the first Harbour Master's Office, of which there is a branch office in Cavtat. Sea samples were obtained at the sampling location along the waterfront always in the same place, at a distance of one meter from the pier.

Table 3 shows that from November to May, no moored vessels were present; as a result, we found no bacteria in the sea samples. Bacteria in the samples significantly increase when more yachts were moored, i.e., when there were more people present. A graphical representation of the results in Table 3 is provided in Figure 5.

**Table 3.** Distribution of vessels and persons on vessels on sampling days at site C, Cavtat waterfront.

| Sampling Point C, Cavtat Waterfront (Yachts and Megayachts) | | | | |
|---|---|---|---|---|
| Sampling Date | Number of Bacteria CFU/100 mL | Type of Vessels at Sampling Location | Number of Vessels | Most Probable No. of Persons |
| 13 Sep. 2012 | 1290 | yachts 20–40 m | 7 | 110 |
| 18 Oct. 2012 | 0 | yachts and boats | 2 | 10 |
| 18 Nov. 2012 | 0 | no vessels | 0 | 0 |
| 20 Dec. 2012 | 0 | no vessels | 0 | 0 |
| 28 Jan. 2013 | 0 | no vessels | 0 | 0 |
| 25 Feb. 2013 | 0 | no vessels | 0 | 0 |
| 24 Mar. 2013 | 4 | sailboat | 1 | 4 |
| 14 Apr. 2013 | 0 | no vessels | 0 | 0 |
| 21 May 2013 | 0 | sailboat | 2 | 10 |
| 26 Jun. 2013 | 28 | yachts 30–40 m | 4 | 45 |
| 15 Jul. 2013 | 145 | yachts 12–40 m | 14 | 138 |
| 16 Aug. 2013 | 1240 | megayachts 60 m and yachts 40 m | 12 | 126 |
| 9 Sep. 2013 | 6 | yachts 12–40 m | 5 | 64 |
| 20 Oct. 2013 | 2 | sailboat | 1 | 4 |

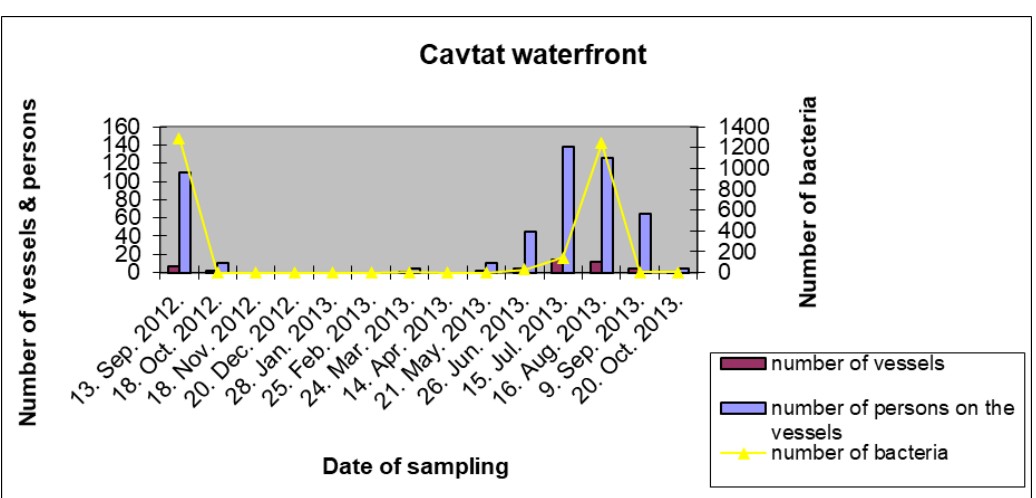

**Figure 5.** Number of fecal coliform bacteria, vessels, and persons on vessels at the sampling location along the waterfront in Cavtat.

3.1.3. Location and Sampling Results in the Gruž Harbour

The port of Gruž is an international harbour for cruise ships and can accommodate up to five large ships with a total number of more than 10,000 passengers. The sampling location was in the immediate vicinity of the first ship in the port (seen from the northwest) and was sampled for 14 consecutive months. (Figure 6.)

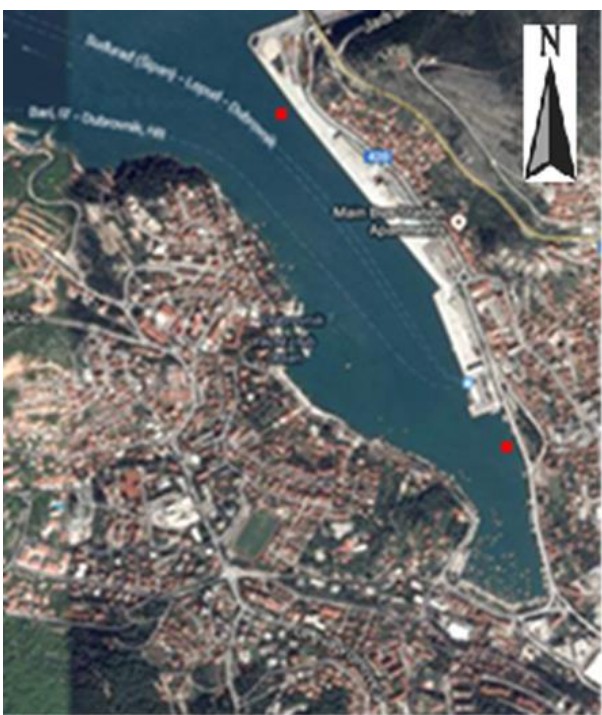

**Figure 6.** Satellite image of Gruž (red dot in the upper part is the sampling point in the harbour, while the red dot in the lower part shows the sampling point along the Gruž waterfront).

The results in Table 4 show that we found no bacteria in the water when cruise ships were berthed in the port. During the four months of the winter, when there were no ships in the port, bacteria was detected in the samples during the months from December 2012 through April 2013 because road reconstruction and installations under the road (from the marketplace to the church and monastery of St. Cross) were occurring. A network of new higher-capacity pipes for rainwater, new sewage pipes, and new power and telephone cable lines was installed. Therefore, at that time, until the completion of the works, the sewage of that part of the city was discharged directly into the sea, which was the cause of the increased number of bacteria in the samples in the port of Gruž along the waterfront. A graphical representation of the results from Table 4 is provided in Figure 7.

**Table 4.** Distribution of vessels and persons on vessels on sampling days at site K, Gruž harbour.

| | Sampling Point K—Gruž Harbour (International Passenger Harbour) | | | |
|---|---|---|---|---|
| **Sampling Date** | **Number of Bacteria CFU/100 mL** | **Type of Vessels at Sampling Location** | **Number of Vessels** | **Most Probable No. of Persons** |
| 12 Sep. 2012 | 0 | cruisers | 3 | 4811 |
| 18 Oct. 2012 | 0 | cruisers | 4 | 5040 |
| 18 Nov. 2012 | 0 | cruisers | 2 | 3257 |
| 20 Dec. 2012 | 78 | no vessels | 0 | 0 |
| 27 Jan. 2013 | 9 | no vessels | 0 | 0 |
| 25 Feb. 2013 | 44 | no vessels | 0 | 0 |
| 24 Mar. 2013 | 239 | cruiser and ro-ro passenger | 2 | 50 |
| 14 Apr. 2013 | 0 | cruisers | 2 | 44 |
| 21 May 2013 | 2 | cruisers | 3 | 5491 |
| 25 Jun. 2013 | 0 | cruisers | 1 | 2422 |
| 14 Jul. 2013 | 0 | cruisers | 4 | 6968 |
| 15 Aug. 2013 | 0 | cruisers | 2 | 1777 |
| 8 Sep. 2013 | 0 | cruisers | 5 | 10,803 |
| 20 Oct. 2013 | 0 | cruisers | 4 | 6604 |

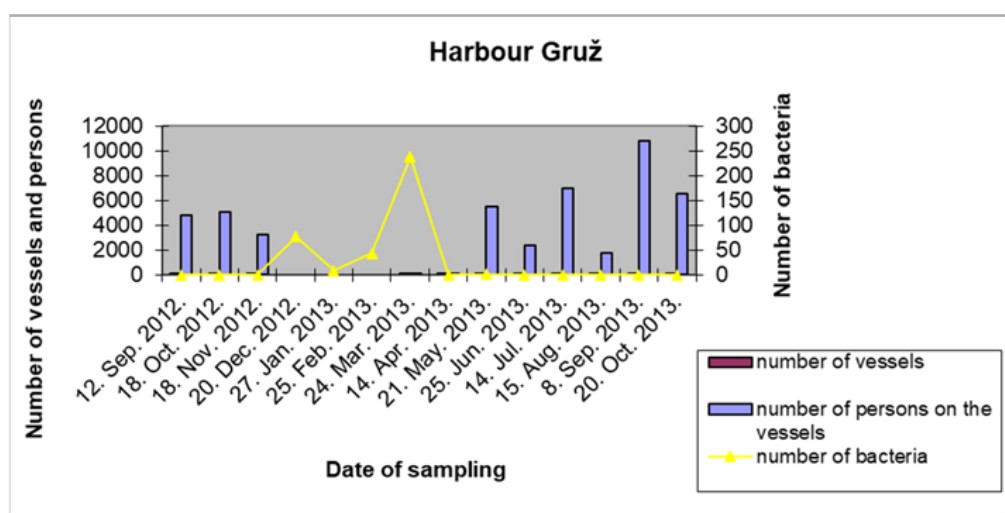

**Figure 7.** Number of fecal coliform bacteria, vessels, and persons on vessels at the sampling location in Gruž harbour.

### 3.1.4. Location and Sampling Results in Gruž along Its Waterfront

Mostly foreign yachts, along with smaller passenger ships in national navigation ("cabotage cruisers"), were moored and resided along the Gruž waterfront. The sampling was conducted along the waterfront across of the middle of Gruž Park, as shown in Figure 4 by a red dot in the lower part of the photograph. The sampling point distance was approximately 30 m from waterfront pier. In Table 5, the results of the sample analysis show the presence of coliform bacteria ranging from 0 to more than 2000. Additionally, the number of vessels, their type, and the best estimated number of persons on vessels varied from month to month or season.

**Table 5.** Distribution of vessels and persons on vessels on sampling days at site P, Gruž waterfront.

| | Sampling Point P, Gruž Waterfront Pier | | | |
|---|---|---|---|---|
| **Sampling Date** | **Number of Bacteria CFU/100 mL** | **Type of Vessels at Sampling Location** | **Number of Vessels** | **Most Probable No. of Persons** |
| 24 Sep. 2012 | 148 | 9 minicruisers, 2 yachts | 11 | 250 |
| 21 Oct. 2012 | 15 | 2 minicruisers | 2 | 26 |
| 18 Nov. 2012 | 308 | minicruiser | 1 | 4 |
| 20 Dec. 2012 | 6 | fishing boat | 1 | 6 |
| 27 Jan. 2013 | 3 | no vessels | 0 | 0 |
| 25 Feb. 2013 | 65 | no vessels | 0 | 0 |
| 24 Mar. 2013 | 1920 | yacht and fishing boat | 2 | 10 |
| 14 Apr. 2013 | 550 | small passenger liner | 1 | 40 |
| 21 May 2013 | 0 | minicruiser | 11 | 154 |
| 25 Jun. 2013 | 2 | minicruiser | 5 | 130 |
| 15 Jul. 2013 | 30 | 9 minicruisers, 5 yachts | 14 | 294 |
| 16 Aug. 2013 | 54 | yachts | 7 | 80 |
| 9 Sep. 2013 | 2060 | 8 minicruisers, 4 yachts | 12 | 212 |
| 20 Oct. 2013 | 0 | minicruiser and yacht | 2 | 32 |

The graphical results from Table 5 are shown in Figure 8.

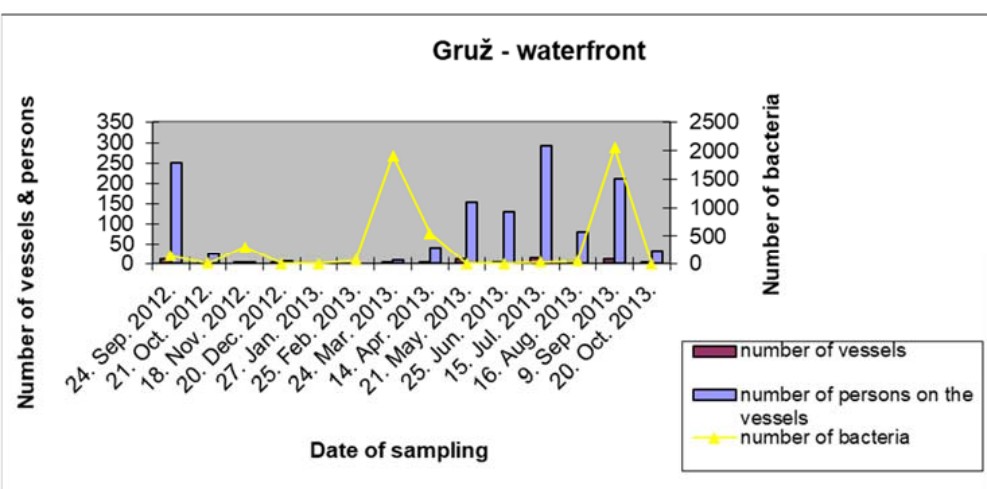

**Figure 8.** Number of fecal coliform bacteria, vessels, and persons on the sampling site in Gruž along its waterfront.

### 3.1.5. Location and Sampling Point at the Anchorage of Lopud Settlement

The Lopud anchorage is an attractive place for boats and yachts during the summer swimming season because it is near beaches, hotels, restaurants, the waterfront, and other facilities of interest to guests of nautical tourism and local boat owners. The sampling location was about 50 m from the beach ashore, to the right, in the middle of the bay, in the direction of a small mole (Figure 9).

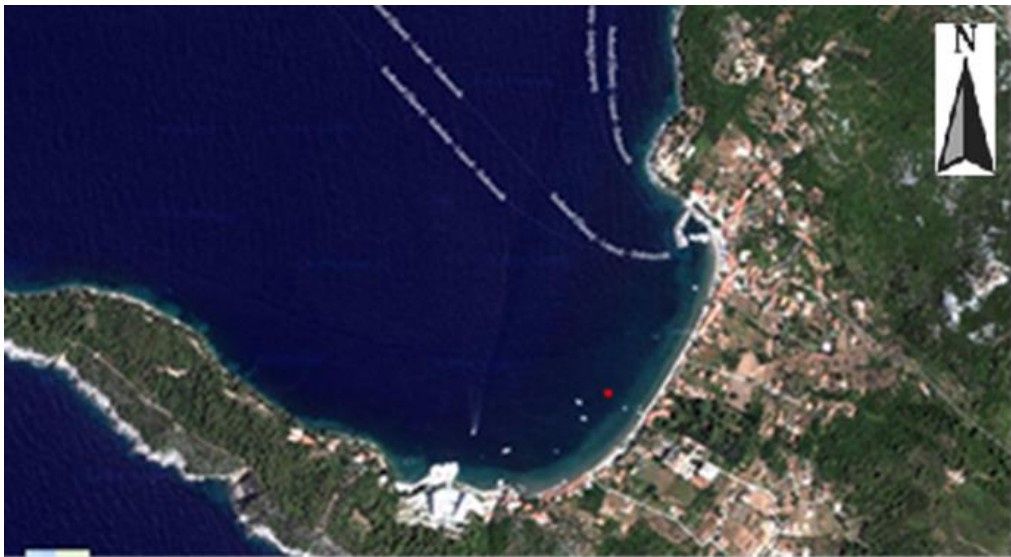

**Figure 9.** Satellite photo of the Lopud anchorage on the island of Lopud (the red dot in the photograph is the sampling site).

The results of the analysis of the sea samples from the Lopud anchorage are shown in Table 6 and demonstrate the presence of fecal coliform bacteria in the samples during the summer months when there were boats and sailboats at the anchorage.

**Table 6.** Distribution of vessels and persons on vessels on sampling days at site L, Lopud.

| | Sampling Point L—Lopud (Inhabited) | | | |
|---|---|---|---|---|
| **Sampling Date** | **Number of Bacteria CFU/100 mL** | **Type of Vessels at Sampling Location** | **Number of Vessels** | **Most Probable No. of Persons** |
| 23 Sep. 2012 | 0 | boats and sailboats 8–10 m | 8 | 24 |
| 21 Oct. 2012 | 0 | no vessels | 0 | 0 |
| 18 Nov. 2012 | 2 | no vessels | 0 | 0 |
| 20 Dec. 2012 | 0 | no vessels | 0 | 0 |
| 27 Jan. 2013 | 0 | no vessels | 0 | 0 |
| 25 Feb. 2013 | 0 | no vessels | 0 | 0 |
| 24 Mar. 2013 | 0 | no vessels | 0 | 0 |
| 14 Apr. 2013 | 0 | no vessels | 0 | 0 |
| 21 May 2013 | 0 | no vessels | 0 | 0 |
| 25 Jun. 2013 | 0 | boats | 1 | 4 |
| 14 Jul. 2013 | 700 | boats and sailboats 8–10 m | 14 | 56 |
| 15 Aug. 2013 | 1760 | boats | 10 | 30 |
| 8 Sep. 2013 | 0 | boats | 4 | 12 |
| 20 Oct. 2013 | 0 | no vessels | 0 | 0 |

The graphical results from Table 6 are shown in Figure 10.

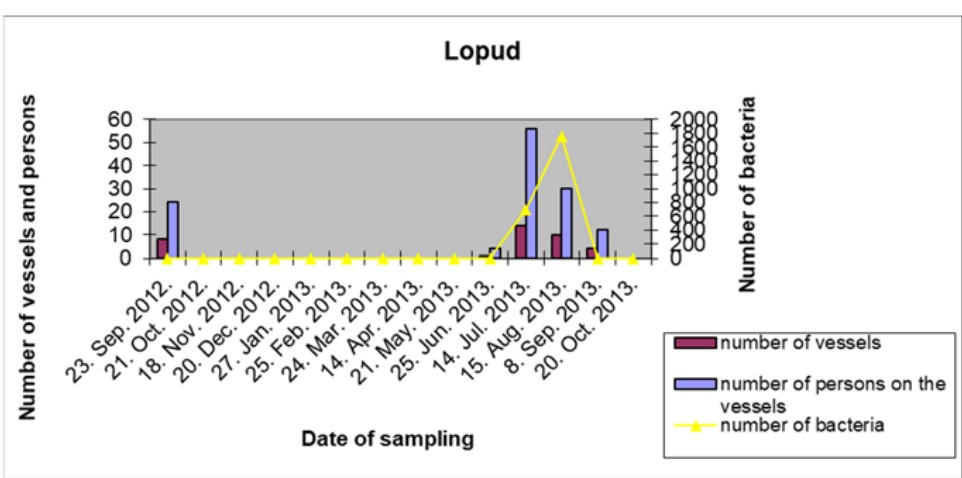

**Figure 10.** Number of fecal coliform bacteria, vessels, and persons on them at the Lopud sampling location.

### 3.1.6. Location and Sampling Results in the Uninhabited Šunj Bay on Lopud Island

Šunj Bay is an uninhabited bay in the south of Lopud Island on the side leeward to the prevailing wind. The beach has fine sand stretches along the entire bay, which is why it is a favourite bathing destination for guests of Lopud, boat owners from Dubrovnik, and other boating tourists. During the swimming season, there can be more than 50 boats anchored in front of the beach during the day, but at night, only a few sailboats or yachts remain. The sampling point was in the middle of the bay closer to the beach, between anchored boats, about 50 m away from the beach shown in Figure 11.

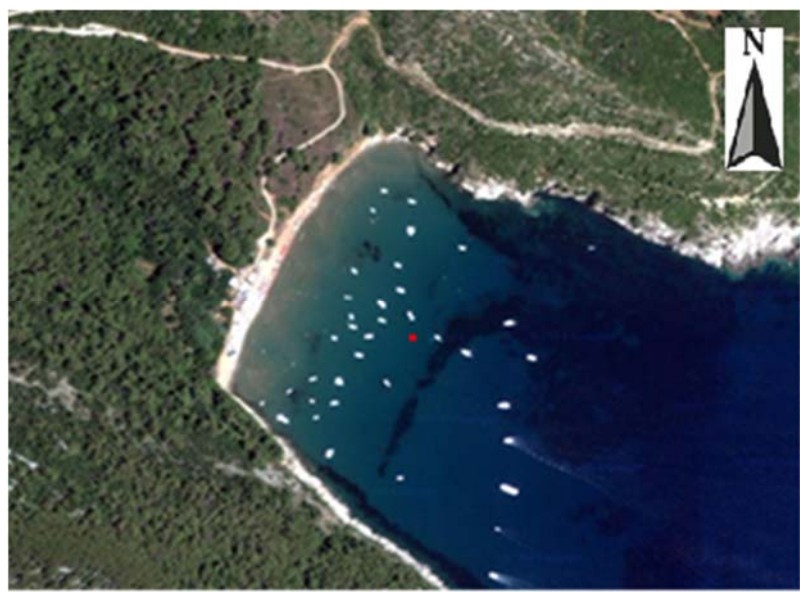

**Figure 11.** A satellite image of Šunj Bay on Lopud Island (the red dot in the photo is the sampling location).

In Table 7, the results of the sea sample analysis show the occurrence, although irregular, of smaller amounts of fecal coliform bacteria in the sea during the summer months when boats, sailboats, and yachts were present at that location.

**Table 7.** Distribution of vessels and persons on vessels on sampling days at site S, Šunj.

| Sampling Date | Number of Bacteria CFU/100 mL | Type of Vessels at Sampling Location | Number of Vessels | Most Probable No. of Persons |
|---|---|---|---|---|
| | | Sampling Point S, Šunj (Uninhabited Bay, near the Beach) | | |
| 23 Sep. 2012 | 0 | boats and sailboats | 6 | 18 |
| 21 Oct. 2012 | 0 | boats and sailboats | 16 | 48 |
| 18 Nov. 2012 | 0 | no vessels | 0 | 0 |
| 20 Dec. 2012 | 0 | no vessels | 0 | 0 |
| 27 Jan. 2013 | 0 | no vessels | 0 | 0 |
| 25 Feb. 2013 | 0 | no vessels | 0 | 0 |
| 24 Mar. 2013 | 0 | no vessels | 0 | 0 |
| 14 Apr. 2013 | 0 | no vessels | 0 | 0 |
| 21 May 2013 | 1 | boats and sailboats | 3 | 12 |
| 25 Jun. 2013 | 0 | 19 boats, 1 yacht | 20 | 67 |
| 14 Jul. 2013 | 74 | 42 boats, 3 yachts | 45 | 156 |
| 15 Aug. 2013 | 0 | 35 boats, 2 yachts | 37 | 125 |
| 8 Sep. 2013 | 31 | boats and sailboats | 33 | 99 |
| 20 Oct. 2013 | 0 | boats and sailboats | 1 | 2 |

The results in Table 7 are graphically represented in Figure 12.

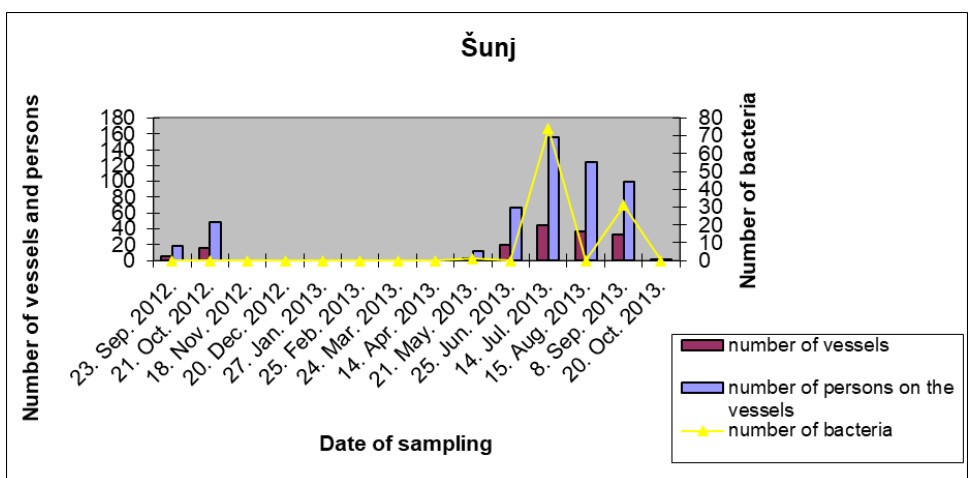

**Figure 12.** Number of fecal coliform bacteria, vessels, and persons on them at the Šunj Bay sampling location.

### 3.1.7. Location and Sampling Results at the Anchorage in Front of Dubrovnik Old Town

The anchorage in front of Dubrovnik old town is a place designated for anchoring cruise ships, megayachts, and yachts. The anchorage is precisely plotted in nautical charts and the cruise ships are always anchored in the same positions. Yachts and megayachts are most often anchored closer to the island. The sampling points (Figure 13.) were selected as close as possible to the vessels themselves (about 10 m from the vessel) because vessels are potential sources of contaminants.

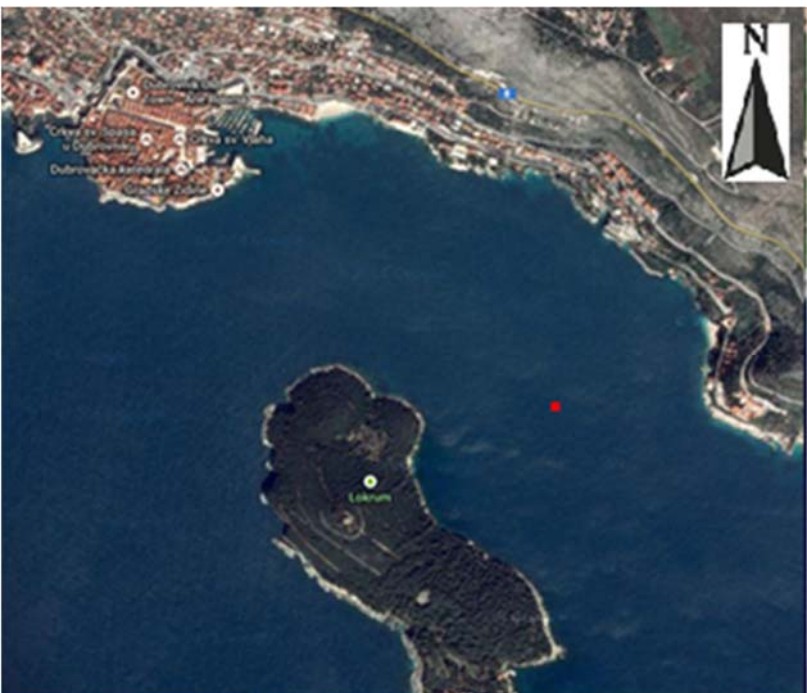

**Figure 13.** Satellite photo of the anchorage of Dubrovnik old town near the island of Lokrum (the red dot in the photo is the sampling site).

The results show that only twice in 14 months did an insignificant amount of bacteria appear at the sampling location.

The graphical results from Table 8 are shown in Figure 14.

**Table 8.** Distribution of vessels and persons onboard vessels on the sampling days at G, Dubrovnik old town anchorage.

| | Sampling Point G—Grad (Dubrovnik Old Town Anchorage) | | | |
|---|---|---|---|---|
| **Sampling Date** | **Number of Bacteria CFU/100 mL** | **Type of Vessels at Sampling Location** | **Number of Vessels** | **Most Probable No. of Persons** |
| 23 Sep. 2012 | 4 | cruiser and yacht | 2 | 3223 |
| 21 Oct. 2012 | 5 | cruiser and yacht | 2 | 3233 |
| 18 Nov. 2012 | 0 | no vessels | 0 | 0 |
| 20 Dec. 2012 | 0 | no vessels | 0 | 0 |
| 27 Jan. 2013 | 0 | no vessels | 0 | 0 |
| 25 Feb. 2013 | 0 | no vessels | 0 | 0 |
| 24 Mar. 2013 | 0 | no vessels | 0 | 0 |
| 14 Apr. 2013 | 0 | cruiser | 1 | 2950 |
| 21 May 2013 | 0 | cruiser | 1 | 382 |
| 25 Jun. 2013 | 0 | cruiser and 2 yachts | 3 | 237 |
| 14 Jul. 2013 | 0 | cruiser | 1 | 2950 |
| 15 Aug. 2013 | 0 | cruiser and yacht | 2 | 398 |
| 8 Sep. 2013 | 0 | megayacht | 4 | 60 |
| 20 Oct. 2013 | 0 | no vessels | 0 | 0 |

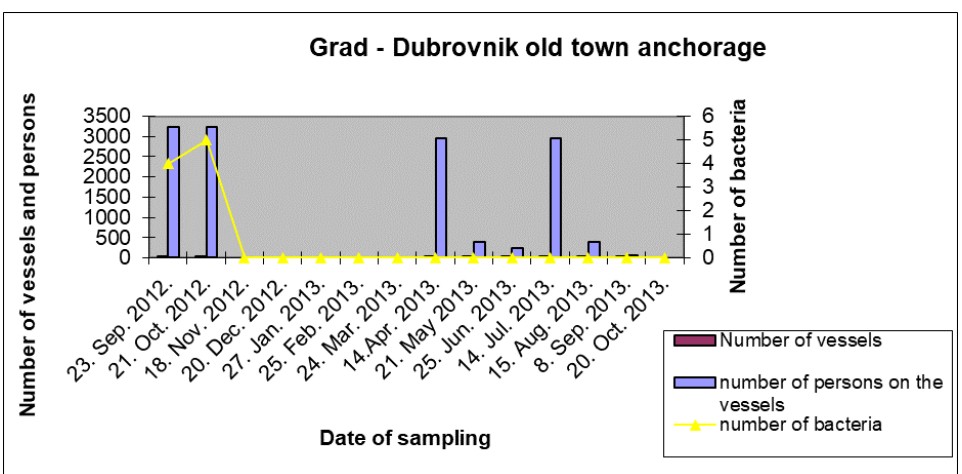

**Figure 14.** Number of fecal coliform bacteria, vessels, and persons at the sampling site at Dubrovnik old town anchorage near the island of Lokrum.

3.1.8. Location and Sampling Results at the Anchorage in Zaton Bay

Zaton Bay has a settlement, so during the summer months at the anchorage in front of the settlement, sailboats, boats, or smaller yachts can anchor in a natural shelter from the prevailing wind or for the night in a sheltered bay. The sampling point was at an anchorage in front of the settlement about 50 m from the shore in the direction of the waterfront shown in Figure 15.

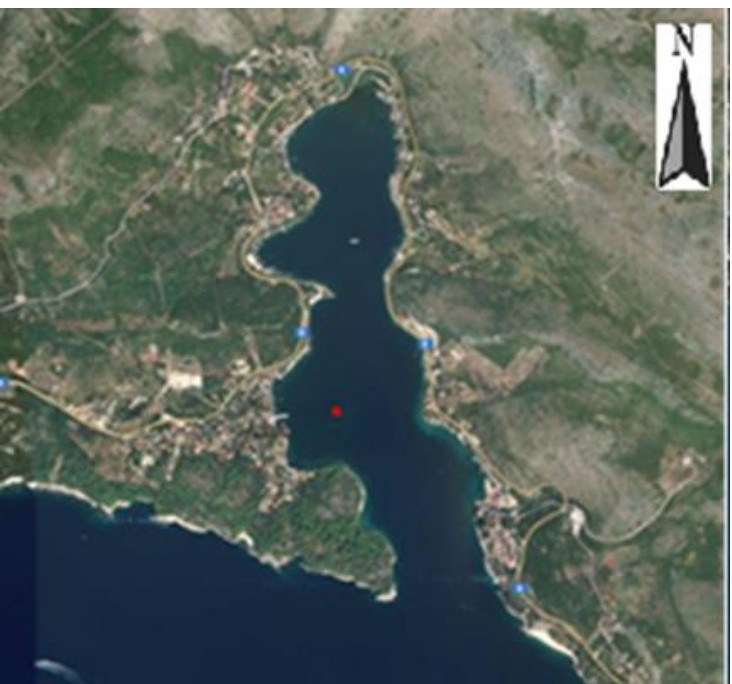

**Figure 15.** Satellite image of the anchorage in front of Zaton (the red dot in the photo is the sampling point).

Vessels were rarely observed at the anchorage in Zaton during sea sampling, even during the summer months. Only once in 14 months did the results of the sample analysis show a smaller amount of fecal coliform bacteria, when a smaller yacht was at anchor. Given the limited number of samples with vessels at anchor at the time of sampling, these results cannot provide a credible basis for conclusions or statements.

Figure 16 shows a graph of the results in Table 9.

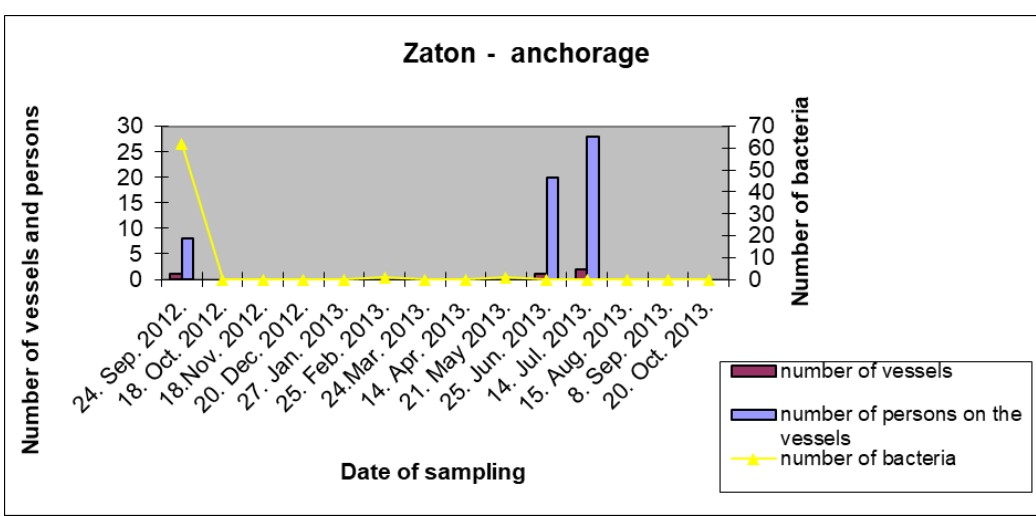

**Figure 16.** Number of fecal coliform bacteria, vessels, and persons on board at the Zaton anchorage sampling location.

**Table 9.** Distribution of vessels and persons on vessels on sampling days at site Z, Zaton.

| | Sampling Point Z, Zaton (Anchorage in front of the Waterfront) | | | |
|---|---|---|---|---|
| **Sampling Date** | **Number of Bacteria CFU/100 mL** | **Type of Vessels at Sampling Location** | **Number of Vessels** | **Most Probable No. of Persons** |
| 24 Sep. 2012 | 62 | yacht | 1 | 8 |
| 18 Oct. 2012 | 0 | no vessels | 0 | 0 |
| 18 Nov. 2012 | 0 | no vessels | 0 | 0 |
| 20 Dec. 2012 | 0 | no vessels | 0 | 0 |
| 27 Jan. 2013 | 0 | no vessels | 0 | 0 |
| 25 Feb. 2013 | 1 | no vessels | 0 | 0 |
| 24 Mar. 2013 | 0 | no vessels | 0 | 0 |
| 14 Apr. 2013 | 0 | no vessels | 0 | 0 |
| 21 May 2013 | 1 | no vessels | 0 | 0 |
| 25 Jun. 2013 | 0 | excursion boat | 1 | 20 |
| 14 Jul. 2013 | 0 | excursion boat | 2 | 28 |
| 15 Aug. 2013 | 0 | no vessels | 0 | 0 |
| 8 Sep. 2013 | 0 | no vessels | 0 | 0 |
| 20 Oct. 2013 | 0 | no vessels | 0 | 0 |

*3.2. Sampling Research and Chi-Square Tests Results*

A chi-square test was performed for each sampling location. The calculation data were obtained from the tables of sample analysis results for each location (Tables 2–9).

These chi-square tests verified the existence of a connection between fecal coliform bacteria in the sea and vessels at the location at the time of sampling. For the chi-square test, the data were obtained on the number and type of vessels at the sampling location and fecal coliform bacteria found in seawater samples. These data, which are quantitative variables, were reduced to their final states for practical application, i.e., to the values of has or does not have, and were observed as qualitative variables. Thus, regardless of the number of vessels at a location, the vessel variable was reduced in response to the following question: Are there any vessels at that location? The answer could be "yes" (if there were more than zero) or "no" (if the number of vessels is zero from the table). Thus, for the bacterial variable, zero bacteria indicated "no", whereas any number above zero implied the value sample contained bacteria. The bacterial variables had their observed (actual) frequency (number of occurrences in samples) and theoretical frequency (theoretical number of times it should appear) with the set null hypothesis. The lowest theoretical frequency was always assumed to have an amount of one because a calculation by formula cannot be performed with zero.

3.2.1. Chi-Square Test for a Group of All Noncruiser Vessels

The chi-square test allows data grouping, so we calculated the first chi-square for a group of all noncruiser vessels at all sampling locations because in the basic scientific hypothesis, smaller and recreational vessels were compared to cruisers (Table 10).

**Table 10.** The chi-square for group of noncruisse vessels.

| The null hypothesis was: | | | |
|---|---|---|---|
| *H0*—Occurrence of fecal coliform bacteria in the sea related to nearby smaller and recreational craft. | | | |

| Frequencies in the table: | | | |
|---|---|---|---|
| Number of samples | Bacteria in the sample | | |
| when there were vessels | has | none | in total |
| Observed frequency * | 35.00 | 16.00 | 51.00 |
| Expected frequencies ** | 38.25 | 12.75 | 51.00 |

* (frequencies from Tables 2, 3, 5–7 and 9)
** Bacteria were expected to appear in the sea where smaller vessels rather than megayachts were located.

| The chi-square was calculated according to Equation (1): | $\chi^2 = \sum \frac{(f_0 - f_t)^2}{f_t}$ | | | using the following table: |
|---|---|---|---|---|
| $f_0$ | $f_t$ | $f_0 - f_t$ | $(f_0 - f_t)^2$ | $\sum \frac{(f_0 - f_t)^2}{f_t}$ |
| 35.00 | 38.25 | −3.25 | 10.56 | 0.28 |
| 16.00 | 12,75 | 3.25 | 10.56 | 0.83 |
| | | | | $\chi^2 = 1.10$ |

| significance level ($\alpha$) | 0.05 | → (determined for 95% probability of the truth of the null hypothesis) |
|---|---|---|
| degree of freedom ($v$) | 1 | → (obtained by calculation) |
| critical value (cv) | 3.84 | → (when $v = 1$ and $\alpha = 0.05$) |
| Result | | |
| chi-square ($\chi^2$) | 1.10 | → Chi-square is less than the critical value. |
| Conclusion | H0 | → Null hypothesis is accepted. |
| | | **The vessels are related to bacteria in the sea.** |

### 3.2.2. Chi-Square Test for a Group of Cruise Ships

Cruisers at two locations (Gruž harbour and an anchorage in front of the Dubrovnik old city) were observed as one group according to the basic scientific hypothesis, so a chi-square test was conducted for that group (Table 11).

**Table 11.** The chi-square for group of cruise ships.

| The null hypothesis was set: | | |
|---|---|---|
| *H0*—The occurrence of fecal coliform bacteria in the sea is related at cruiser locations. The chi-square according to Equation (1) was calculated as: $\chi^2 = \sum \frac{(f_0 - f_t)^2}{f_t}$ where we used the observed frequencies from Tables 4 and 8. For the expected frequencies, we expected bacteria to appear in the sea where cruisers were located. | | |
| significance level ($\alpha$) | 0.05 | → (determined for 95% probability in the truth of the null hypothesis) |
| degree of freedom ($v$) | 1 | → (obtained by calculation) |
| critical value (cv) | 3.84 | → (when v = 1 and $\alpha = 0.05$) |
| Result | | |
| chi-square ($\chi^2$) | 9.53 | → The chi-square is greater than the critical value. |
| Conclusion | H0 | → The null hypothesis was rejected. |
| | | **The vessels are not related to bacteria in the sea.** |

### 3.2.3. Chi-Square Test for the Sampling Locations

After the chi-square tests were performed for each sampling location, the obtained results are shown in Tables 12 and 13.

**Table 12.** Table of vessel correlation with fecal coliform bacteria at the sampling locations.

| | Chi-Square Test for Sampling Locations | | | |
|---|---|---|---|---|
| Sampling Location | (cv) Critical Value | ($\chi^2$) Chi-Square | H0 Null Hypothesis | Vessels and Bacteria Correlation Results |
| ACI Marina Dubrovnik | 3.84 | 1.10 | accepted | correlation exists |
| Waterfront in Cavtat | 3.84 | 1.13 | accepted | correlation exists |
| Gruž harbour | 3.84 | 70.40 | rejected | does not exist |
| Waterfront in Gruž | 3.84 | 1.09 | accepted | correlation exists |
| Anchorage in Lopud | 3.84 | 0.20 | accepted | correlation exists |
| Šunj Bay | 3.84 | 0.50 | accepted | correlation exists |
| Anchorage in front of the old city | 3.84 | 28.57 | rejected | does not exist |
| Anchorage in Zaton | 3.84 | 1.50 | accepted | correlation exists |

**Table 13.** Table of type of vessel and correlation with fecal coliform bacteria at the sampling locations.

| Location | Vessels | Correlation |
|---|---|---|
| All locations in groups | recreational boats (group) | existing |
| ACI Marina | boats, yachts, and megayachts | existing |
| Cavtat waterfront | boats and yachts | existing |
| Šunj Bay | boats, sailing boats, and yachts | existing |
| Lopud Bay | boats and sailing boats | existing |
| Zaton anchorage | yachts | existing |
| Gruž waterfront pier | mini cruisers and yachts | existing |
| Harbour and anchorage | **cruisers** (group) | **does not exist** |
| Dub. old town anchorage | **cruisers** | **does not exist** |
| Gruž harbour | **cruisers** | **does not exist** |

Table 13 shows the following:

1.   A correlation existed between certain types of vessels (boats, yachts, megayachts, sailboats, and smaller cruisers in national navigation) and fecal coliform bacteria in the sea at multiple sampling locations where these vessels were located;
2.   No correlation was found between cruisers and bacteria at both sampling sites where they were located.

The results of sea sampling and data processing by chi-square tests unequivocally confirmed that the risk of black wastewater marine pollution along the coast was higher and less acceptable during the navigation and accommodation of smaller and recreational vessels than for large cruise ships.

From Table 12, which shows the results of the chi-square test for seawater samples, the following is evident:

1.   Cruisers belong to the group posing the lowest level of risk of marine pollution by sewage (black wastewater), and no connection was found between them and fecal coliform bacteria in the sea.
2.   Smaller recreational and commercial vessels (boats, yachts, sailboats, and smaller cruisers in national navigation) belonged to the group of vessels posing the highest risk of marine pollution by black wastewater, showing a connection with bacteria in the sea at several sampling sites.

## 4. Conclusions

In this study, we found that increased traffic of different vessels in the Dubrovnik aquatorium during the summer and tourist season influence coastal sea water contamination with sewage from vessels. This study's results showed that different types of vessels had correlations with sewage pollution in certain coastal areas. The research results proved that smaller and recreational vessels are responsible for sewage pollution of the coastal sea in the researched area. Additionally, large cruise ships were not correlated with sewage in harbours and anchorages where they were located. The methods and results of this research can find practical application and serve as a basis for further research related to the pollution of coastal seas with black waste water discharged from vessels because the method can be applied to any other coastal water area worldwide. Likewise, these results can help with the adoption of new national or international legal regulations that will more effectively protect the sea along the coast from pollution caused by sewage (black wastewater) discharged from all type of vessels, including small and recreational vessels, and thus better protect the economy based on tourism and ensure sustainable economic growth and development of maritime transport.

**Author Contributions:** Data curation, R.C.H.; Visualization, N.K.; Writing–original draft, Ž.K.; Writing–review & editing, D.M. All authors have read and agreed to the published version of the manuscript.

**Funding:** This research received no external founding.

**Institutional Review Board Statement:** Not applicable.

**Informed Consent Statement:** Not applicable.

**Data Availability Statement:** Data available on request.

**Conflicts of Interest:** The authors declare no conflict of interest.

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
