# Peer review of "Analysis of Sea Pollution by Sewage from Vessels"

_sustainability, doi:10.3390/su14010263_

Round 1

Reviewer 1 Report

Introduction:

Form me it is not clear why you perform the study. There is a official and regular monitoring of bacterial contamination by agencies/offices, right? So why are you doing the monitoring? And are there any laws or provisions for releasing untreated wastewater? Do you have any evidence that boats do release untreated wastewater?

Material and Methods:

Line 150: reference format

Do you know how many boats do release their untreated wastewater to the seawater? If I understood correctly they have to use specific treatment infrastructure and it is not allowed to release untreated wastewater. Do you know if any of the vessels did it?

Your tests for correlations is questionable as it depends on how many vessels do release untreated wastewater to the seawater. If there are 10 vessels with 50 persons that follow the rules and use treatment infrastructure and on the other hand you have 3 boats with 9 persons that do not care and release all their untreated wastewater, then your correlation does not work with the number of vessels and persons.

Did you use a negative control (seawater from an area, where no or only very few vessels are)?

What is with coliform bacteria from other sources, eg marine animals? How do you know that the coliform bacteria are originated from the vessels?

You described the methods and the material used in detail, but you did not mention how the samples were analyzed. What filters were used? What plates for cultivation? etc. You do not need to explain how you sterilize your equippment.

Results and Discussion:

I like that you divided this section by sampling sites.

Lines 171-174: What is with temperature? Maybe you could not detect coliform bacteria, because in the winter months, when there are no vessels the water temperature i slow, maybe too low for those fecal bactetria?

Table 1: Is it CFU/mL?

Line 184: Delete „This Marina is located“

Table 2: Again: please give the unit for number of bacteria

Lines 240-242: please correct sentence

Lines 352-358: I do not think that it is necessary to describe the principle of statistical tests. This is unusual. Just describe the test and its parameters in the Material and Methods part and mention the results here. You do not need to describe what significance or degree of freedom is…

Reviewer 2 Report

Koboevic et al- Analysis of Sea Pollution by Sewage from Vessels

Line 9- Need to improve and need to be more specific. Coastal and sea are redundant-

Line 10- Is it a laboratory experiment or did you sample from the coastal area?

Line 12- What is faecal marine pollution? Is faecal pollution not sufficient?

Line 13- “Obtained results….” this sentence is not needed.

  • THE WRITING NEEDS TO BE IMPROVED, CREATE A STORY, MORE CONCISE AND BETTER ENGLISH.
  • NEED TO CLEAR FROM THE BEGINNING, WHAT YOU ARE SAYING-
  • What do you want to focus on- marine pollution due to vessels? Direct your writing always there.

NEED TO READ MORE PAPERS RELATED TO FECAL POLLUTIONS:

(Ananda Tiwari et al., 2021)

(Shrestha et al., 2020)

(Meals et al., 2013)

(A. Tiwari et al., 2016)

(Ananda Tiwari et al., 2018)

REFERENCES

Meals, D. W., Harcum, J. B., & Dressing, S. A. (2013). Monitoring for Microbial Pathogens and Indicators. TechNotes, 1–29. http://www.bae.ncsu.edu/programs/extension/wqg/319monitoring/tech_notes.htm

Shrestha, A., Kelty, C. A., Sivaganesan, M., Shanks, O. C., & Dorevitch, S. (2020). Fecal pollution source characterization at non-point source impacted beaches under dry and wet weather conditions. Water Research, 182. https://doi.org/10.1016/j.watres.2020.116014

Tiwari, A., Niemelä, S. I., Vepsäläinen, A., Rapala, J., Kalso, S., & Pitkänen, T. (2016). Comparison of Colilert-18 with miniaturised most probable number method for monitoring of Escherichia coli in bathing water. Journal of Water and Health, 14(1). https://doi.org/10.2166/wh.2015.071

Tiwari, Ananda, Hokajärvi, A. M., Santo Domingo, J. W., Kauppinen, A., Elk, M., Ryu, H., Jayaprakash, B., Pitkänen, T., & Domingo, J. W. S. (2018). Categorical performance characteristics of method ISO 7899-2 and indicator value of intestinal enterococci for bathing water quality monitoring. Journal of Water and Health, 16(5), 711–723. https://doi.org/10.2166/wh.2018.293

Tiwari, Ananda, Oliver, D. M., Bivins, A., Sherchan, S. P., & Pitkänen, T. (2021). Bathing Water Quality Monitoring Practices in Europe and the United States. International Journal of Environmental Research and Public Health, 18(11), 5513. https://doi.org/10.3390/ijerph18115513

Round 2

Reviewer 2 Report

Now, the quality of the manuscript has improved than the earlier version. But still, it is not sufficient. Please can you send the manuscript for English language correction?  

Author Response

Dear Reviewer,

According to your advice the manuscript has been sent for English language correction. MDPI English Editing Service sent to me edited version back, so that I uploaded right now to the system.  Many thanks for your help. I am taking this opportunity to wish you Merry Christmas and Happy new Year.

Kind regards.